# Comparison of Rabbit, Kitten and Mammal Milk Replacer Efficiencies in Early Weaning Rabbits

**DOI:** 10.3390/ani10061087

**Published:** 2020-06-23

**Authors:** Panthiphaporn Chankuang, Achira Linlawan, Kawisara Junda, Chittikan Kuditthalerd, Tuksaorn Suwanprateep, Attawit Kovitvadhi, Pipatpong Chundang, Pornchai Sanyathitiseree, Chaowaphan Yinharnmingmongkol

**Affiliations:** 1Faculty of Veterinary Medicine, Kasetsart University, Bangkok 10900, Thailand; panthiphaporn.c@ku.th (P.C.); achira.l@ku.th (A.L.); kawisara.ju@ku.th (K.J.); chittikan.k@ku.th (C.K.); tuksaorn.s@ku.th (T.S.); 2Department of Physiology, Faculty of Veterinary Medicine, Kasetsart University, Bangkok 10900, Thailand; pichandang@gmail.com; 3Department of Large Animal and Wildlife Clinical Science, Faculty of Veterinary Medicine, Kasetsart University, Nakorn Pathom 73140, Thailand; fvetpos@ku.ac.th; 4Animal Space Pet Hospital, Bangkok 10170, Thailand; taaoy@hotmail.com

**Keywords:** digestibility, enzyme activity, gut histology, milk replacer, rabbit

## Abstract

**Simple Summary:**

A milk replacer must be given as the main diet to young rabbits that are separated from their mothers before they reach weaning age (31–35 days). This procedure, which is a rescue protocol, allows them to survive. Moreover, the early separation of young rabbits before weaning prevents negative consequences in lactating rabbits, which is beneficial to pet rabbit producers. Kitten (KMR^®^, Pet-Ag Inc., Hampshire, IL, USA: KMR) or mammal (Zoologic^®^ Milk matrix 30/52, Pet-Ag Inc., Hampshire, IL, USA: MMR) milk replacers have generally been suggested for use in rabbits; however, rabbit milk has a unique composition. Therefore, a rabbit milk replacer (RMR) was formulated in this study for comparison with these commercial products. Early weaned rabbits at 18 days of age were fed daily with RMR, KMR or MMR until 36 days after birth, while a commercial pelleted diet and water were provided at an amount exceeding the normal intake. The results indicated that it is possible to use RMR as a milk replacer for rabbits without serious adverse consequences. However, the RMR group presented a lower trend in nutrient digestibility than the other groups, although there was no statistical significant difference. Therefore, prebiotics and/or probiotics should be added to RMR formulations to improve this parameter.

**Abstract:**

Early weaned rabbits should be fed using a milk replacer in order to survive. Therefore, a rabbit milk replacer (RMR) was developed and compared with a kitten milk replacer (KMR^®^: KMR) and a mammal milk replacer (Zoologic^®^ Milk matrix 30/52: MMR). Thirty-six native crossbred rabbits aged 18 days were divided into three experimental groups (six replicates/group, two rabbits/replicate), fed RMR, KMR or MMR daily until they were 36 days old and euthanized at 38 days, while a complete pelleted diet and water were provided *ad libitum*. No statistically significant differences were observed in growth performance parameters, water intake, faecal weight, nutrient digestibility, internal organ weight, caecal pH, caecal cellulose activity, number of faecal pellets and amount of crude protein intake (*p* > 0.05). Caecal amylase activity in the KMR group and caecal protease activity in the RMR group were higher than in the MMR group (*p* < 0.05). The villus height and crypt depth of the MMR group were greater than in the RMR and KMR group (*p* < 0.05). In conclusion, it is possible to feed RMR to early weaning rabbits without serious adverse effects. However, probiotics and/or prebiotics should be supplemented in milk replacers and their benefits studied.

## 1. Introduction

The size of the pet market has increased sharply in recent years and was estimated to be around 131.7 billion US dollars in 2016 [1]. Moreover, the compounded annual growth rate of the global pet care market was forecast to be 4.9% between 2018 and 2025 due to changes in the new generation’s lifestyle, such as living alone or child-free marriage [1]. Nevertheless, humans still need interactions with living things, which add social, medical, emotional and physical benefits to their lives [2]. Companion animals are one solution that can offer these benefits [2]. Although rabbits are not as popular as dogs and cats, they occupy third place among companion animals because they are clean, quiet, non-harmful and require little space [3].

Generally, rabbits are weaned at 31–35 days of age by rabbit producers in Thailand and other countries [4,5] and begin to consume pelleted diets around 18 days of age [6]. Milk replacer has been suggested as a means of feeding early weaned rabbits and orphaned rabbits and of solving the problem of female rabbits that do not produce milk, this being rabbits’ major nutrient source for survival [7] as well as preventing gastrointestinal disease [8]. The early separation of young rabbits from their mother can prevent a negative energy balance due to lactation, which supports a higher production yield and reduces disease transmission from the mother to young rabbits, of benefit for pet rabbit producers [4,9]. In addition, the smallest rabbit breed (Netherland Dwarf) can produce around 4–6 kittens per litter; however, the milk yield of this breed is not sufficient to support their kittens, leading to a high mortality rate among young rabbits [5]. Furthermore, milk replacer can be used to rescue unweaned wild rabbits [8]. Therefore, the use of a milk replacer provides one solution to these problems.

Commercial rabbit milk replacer remains lacking or unavailable in some countries. For this reason, kitten milk replacer (KMR) has been suggested as a substitute [7]. However, although kitten milk replacer can be used in early weaning rabbits, its growth performance has been found to be inferior to rabbit milk for rabbits [8]. In addition, two milk replacer formulas using mixtures of kitten milk replacers—Fox Valley Ultraboost and/or Fox Valley 32/40 (Fox Valley Animal Nutrition, INC., Lakemoor, IL, USA)—have been used to rescue young desert cottontail (*Sylvilagus audubonii*) and eastern cottontail rabbits (*S. floridanus*), for which the mortality rate was 26–59% [8]. The high concentration of nutrients (fat, protein and energy), the near absence of lactose, the high proportion of medium-chain saturated fatty acids with bacteriostatic properties (C8:0 and C10:0) and the short milking period required constitute the unique characteristics of rabbit milk and feeding behaviour [6,10]. Therefore, milk replacer for rabbits should be formulated respecting the properties of real rabbit milk [10]. Moreover, cost-effectiveness is another problem for rabbit producers, owners and wildlife rescue center [8]. Therefore, this study aimed to compare the efficiency of a developed rabbit milk replacer with two commercial products (kitten and mammal milk replacers) based on the growth performance and health status of early weaning rabbits (18 days old).

## 2. Materials and Methods 

### 2.1. Ethics Statement

This study was conducted following standard guidelines at the animal experimental unit, Faculty of Veterinary Medicine (Kasetsart University, Bangkok, Thailand) and was approved by the Institutional Animal Care and Use Committee of Kasetsart University, Bangkok, Thailand (ACKU62-VET-037).

### 2.2. Animals, Diets, Milk Replacer Preparation and Experimental Design 

Thirty-six 18-day-old native crossbreed rabbits with initial body weights of 134 ± 6.31 g/head (mean ± standard deviation) were taken from a local rabbit farm (Saha farm, Kanchanaburi, Thailand). Rabbits were randomly separated into three experimental groups with equal numbers of each sex (six replicates per group and two rabbits per replicate) containing: (1) rabbits fed rabbit milk replacer, which was formulated in this study (RMR); (2) rabbits fed kitten milk replacer (KMR^®^; Pet-Ag Inc., Hampshire, IL, USA); and (3) rabbits fed mammal milk replacer (Zoologic^®^ Milk matrix 30/52; Pet-Ag Inc., Hampshire, IL, USA; MMR). Rabbits were placed in a stainless cage (35 cm × 35 cm × 35 cm) with controlled room temperature, light and humidity at 20 ± 2 °C, 16L:8D and 75 ± 10%, respectively. The experiment was conducted for 20 days until the rabbits reached 38 days of age. At the end of the experiments, one rabbit per replicate was euthanized by intraperitoneal injection with pentobarbital sodium at 100 mg/kg (Nembutal, Ceva corporate, France) [11] and the samples were collected for further analysis, while another rabbit of each replicate was returned to the farm and reared until it reached 60 days of age.

The formulation of the RMR, including the chemical composition of the milk replacer, complete pelleted diet and rabbit milk, is illustrated in Table 1. Every day, rabbits were fed 10 mL at 38 °C of a freshly prepared mixture of milk replacer powder and clean water using a sterile syringe at 7:00 until they reached weaning age (36 days), while clean water and complete commercial rabbit diets (Lee Feed Mill, Publ. Co., Ltd., Phetchaburi, Thailand) were provided *ad libitum* throughout the experiment. The KMR and MMR powders were diluted with warmed water at a 7:13 ratio and homogenized. The RMR consisted of two parts: a hydrogenated palm fat part and a mixed dried powder part, containing all ingredients except hydrogenated palm fat and polyoxyethylene (80) sorbitan monooleate. For RMR preparation, hydrogenated palm fat was heated in an 800 W microwave for 60 s, changing from solid to liquid form as a result. The dried mixed powder part was then mixed with warm water. Subsequently, the two parts were homogenized and polyoxyethylene (80) sorbitan monooleate was added as an emulsifier. The ratio of mixed dried powder to hydrogenated palm fat to warm water was 24.5:17.5:78. The dilution ratio was selected based on equal dry matter content between the milk replacers and the solubility of the milk mixture.

### 2.3. Perfomance, Digestibility and Faecal Evaluation

The animals were weighed at 18, 24, 30 and 36 days of age, whereas average daily feed intake (ADFI), average daily weight gain (ADG), feed conversion ratio (FCR), water intake and weight of faeces output were evaluated at 19–24, 25–30 and 31–36 days of age. The apparent digestibility of dry matter, organic matter, ether extract and crude protein was conducted at 23–27 and 31–35 days of age and contained six replicates/groups. The procedures for feeding, faecal collection, chemical analysis and calculation were in accordance with [14]. Briefly, feed intake was measured during the period of the digestibility trial. Faeces were removed from the cage at 9:00 on the first day of the digestibility trial. Subsequently, all faeces on a net under the cage were collected at 9:00 for four days. The faeces were weighted immediately after collection, put in a sterile plastic bag and kept at −20 °C for further chemical composition analysis following the procedure of [14]. Another study was conducted, where the amount of daily faecal pellet excretion was measured by counting the dried faecal pellets between 19 and 36 days old from photos.

### 2.4. Internal Organs, Gut Histology and Caecal pH

The internal organ weight and the body weight of the euthanized rabbits were determined. The duodenal part of the small intestine was fixed in 10% buffered formalin for further villus morphometric evaluation. Briefly, small pieces of middle duodenum after fixation were processed, embedded in paraffin, sectioned at 7-µm thicknesses by means of a rotary microtome (Leica RM2155; Leica Instruments GmbH, Nussloch, Germany) and stained by haematoxylin and eosin method. Villi height and crypt depth were evaluated under a microscope using an image analysis programme (Image Pro Plus; Media Cybernetics, Bethesda, MD, USA). Caecal pH was measured directly using a Crison MicropH 2001 pH meter (Crison Instruments, Barcelona, Spain). The caecal content was immediately placed in sterile plastic tubes under ice for enzyme preservation and kept at −20 °C for further analysis of caecal enzyme activity. 

### 2.5. Caecal Enzyme Activity

The crude enzyme extracted from the caecal content was extracted by homogenized caecal content with phosphate buffer solution (pH 7) at a 1:5 ratio (*w*/*v*). The homogenates were centrifuged at 18,000× *g* for 30 min at 4 °C to obtain the supernatant used to evaluate amylase, protease and cellulase activity. Amylase and cellulase activity were assayed according to [15,16] using 5% soluble starch and 1% carboxyl-methyl cellulose (CMC; medium viscosity) as the substrate, respectively. One hundred microlitres of crude enzyme extract were added to activate the digestion of the substrates. The products of the carbohydrate-digestive enzymes were stained using 1% dinitrosalicylic acid (DNS) and measured using a spectrophotometer at 540 nm against a linear range of maltose standards for amylase and glucose standards for cellulase. Protease activity was assayed according to the method described by [17] using 0.6% casein as the substrate. The product of the protein-digesting enzyme was measured spectrophotometrically at 660 nm against a linear range of tyrosine. The activity of the observed digestive enzymes was expressed as U. 

### 2.6. Crude Protein Assessment

Each rabbit’s feed intake between 19–24, 25–30 and 31–36 days old and the amount of crude protein in the milk replacer and diet were used as information to calculate the amount of crude protein intake.

### 2.7. Statistical Analysis

The results of this study are represented as the mean and pooled standard error of the mean. A completely randomized design was employed in this study. Therefore, one-way analysis of variance (ANOVA) was used to compare the different types of milk replacers (fixed factors) for internal organ characteristics, caecal pH, caecal digestive enzyme activities and duodenal histology, whereas the growth performances, water intake, faeces excretion, apparent digestibility, number of faecal pellets and amount of protein intake were analyzed by two-way mixed analysis of variance, with treatment groups or age serving as the between-subjects or the within-subjects factor, respectively. Duncan’s multiple range test was used for post hoc analysis. Differences were considered statistically significant at *p* < 0.05. All statistical analyses in this study were performed with R-statistic software using the Rcmdr package [18].

## 3. Results

The effects on performance, apparent digestibility, amount of faeces excretion and crude protein intake from rabbits fed the different milk replacers are shown in Table 2. No statistically significant differences between the groups and the interactions between the studied factors (groups and age) for all parameters in Table 2 were apparent (*p* > 0.05). The age increment was correlated with increased body weight, ADFI, ADG, FCR, water intake, faeces excretion and crude protein intake (*p* < 0.05), whereas apparent digestibility did not affect dry matter, organic matter or ether extract (*p* > 0.05). However, the crude protein digestibility of rabbits at 31–35 days old was lower than rabbits at 23–27 days old (*p* < 0.05). The rabbits fed KMR and MMR displayed higher nutrient digestibility in both age ranges compared with rabbits fed RMR, but there was no statistically significant difference (*p* > 0.05). Rabbits in RMR, KMR and MMR were received the crude protein from milk daily at 2.23, 2.61 and 1.62 g dry matter/head. No deaths, morbidities or clinical signs were observed in rabbits during the experimental period. Moreover, a rabbit in each replicate was not euthanized at the end of the experiment and remained alive until it reached two months of age. In addition, there were no problems of milk perception and palatability in any group in this experiment, because the rabbits sucked milk directly from the syringe without any force feeding.

The consequences for internal organ weight, caecal pH, duodenal wall histology and digestive enzyme activities between the groups are compared in Table 3. The weight of each internal organ was not statistically significantly different between the groups (*p* > 0.05). Caecal pH was not influenced by the differences in milk replacers (*p* > 0.05). Caecal amylase activity in the MMR group was lower than in the KMR group (*p* < 0.05), whereas greater caecal protease activity was observed in the RMR group compared to the MMR group (*p* < 0.05). Cellulase activity was not affected by the treatments (*p* > 0.05). Respectively, the shortest villus and the shallowest crypt depth were found in the RMR and the KMR groups compared to the MMR group (*p* < 0.05).

The average number of faeces pellets from two rabbits of each replicate in the experiments is illustrated in Figure 1 and Table A1. The graph shows a steady increase in the number of faecal pellets with increasing age (*p* < 0.001); however, a sharp drop occurred in all study groups at 35 days of age, followed by another increase. The largest number of faecal pellets existed in the MMR group compared to the RMR group (*p* < 0.05; Appendix A
Table A1), with the KMR group between them (*p* > 0.05). A significant interaction between the fixed factors (age and treatment group) was identified (*p* < 0.05). Generally, the same increasing trend was observed in all groups, except for the sharp rise in the number of faecal pellets in the MMR, RMR and KMR groups at 22–23, 24–26 and 34–36 days of age, respectively.

## 4. Discussion

The RMR was formulated according to the profile of rabbit milk; therefore, its chemical composition was the most similar to the composition of rabbit milk [10]. KMR contained a higher proportion of crude protein and was lower in fat and energy than RMR and rabbit milk. On the other hand, MMR was lower in crude protein and higher in fat and energy than RMR and rabbit milk. A high density of nutrients and energy was the unique characteristic of rabbit milk, which contained respectively around four and three times higher proportions of protein and lipids than cow’s milk [10]. A short milking time is a common nursing behaviour, explaining the high density of rabbit milk [6]. The amount of milk replacer fed to young rabbits was calculated on the basis of stomach capacity. Milk replacer in powder form was used for all formulations in this study because a highly concentrated milk mixture can be formulated from dried powder but not in liquid form [8,9]. Rabbit milk protein comprises around 70% and 30% casein and whey protein, respectively [10]. Therefore, casein served as a major protein ingredient in the milk replacer formulation for RMR and the two other commercial milk replacers, whereas dried skimmed milk powder represented another protein source, which was used in a lower proportion than casein. Respectively, either whey or milk protein concentrate was supplemented in KMR and MMR, whereas in RMR they were not. A low level of lactose is present in rabbit milk; therefore, cow’s or goat’s milk is limited as a milk replacer formulation [10]. Moreover, lactase activity in rabbits decreases with age and does not respond to the lactose concentration in the diet. Therefore, excessive lactose intake can lead to a digestive disorder [6]. Differences in the type of raw protein source and quantity influence the diversity of the amino acid profile. Although the amino acid profile was not evaluated in this study, the intake of amino acids would have been sufficient for rabbits because casein, considered an ideal protein, was used as the main ingredient and the percentage of crude protein in all milk replacers was higher than nutrient requirements [4,6].

Fat in rabbit milk represents the major energy sources for rabbits [10], whereas excessive starch intake promotes digestive problems and increases the mortality rate of young rabbits [4]. The highest nitrogen-free extract was present in the KMR formulation; however, no adverse effects were observed in this group. Medium-chain fatty acids, mainly caprylic (C8:0) and capric acid (C10:0), were the major components of fatty acids in rabbit milk, comprising around 50% of total fatty acids [10]. Vegetable oil or hydrogenated palm oil served as the only lipid sources in milk replacers, comprising a high proportion of polyunsaturated fatty acids; however, they still contained these medium-chain fatty acids. Another function of caprylic and capric acid was their antibacterial properties, which maintained microbial community development and prevented pathogen invasion. Supplementation with these medium-chain fatty acids should confer health benefits on rabbits. Nevertheless, no health problems occurred in this study, although these medium-chain fatty acids were not supplied in the formulations. The essential fatty acids were supplied in sufficient amounts to fulfil the nutritional requirements of all the study groups, as there was a very high amount of fat in all the formulations [6]. However, an antioxidant such as tocopherol should be supplemented to prevent lipid oxidation. The minerals and vitamins in RMR were higher than the minimum requirements in rabbits [6]. In the early period of the experiment, the rabbits received nutrients and energy from the milk replacers in high proportions compared to the pelleted diet, which represented only around 1/3 of the total intake in the first period. Subsequently, a higher intake of the pelleted diet became the main nutrient and energy source. Growth performances did not differ between treatment groups, and no morbidity or mortality of rabbits occurred during or after the experimental period. Moreover, the rabbits’ final body weights were similar to those of their counterparts reared with their mother at the same age, as reported in the study of [5]. These findings demonstrate the potential of using milk replacers in early weaned rabbits. However, the low energy and nutrient requirements of the rabbits in this study were characteristics of a native breed with a slow growth rate compared to hybrid or commercial meat rabbits [5]. Therefore, milk replacers can certainly be used in local breed or pet rabbits, possibly offering benefits to veterinarian and pet rabbit producers. Rabbits raised in intensive rearing systems and/or with high growth rates may be studied further as to which milk replacer can support their growth performances and the maintenance of a healthy condition. Another study [9] reported lower growth performances in commercial meat rabbits fed KMR compared with rabbits fed rabbit milk.

A previous report has shown that the gut microbial community of rabbits plays an important role relating to the efficiency of nutrient fermentation, productive performance and heath condition [19]. A sterile gut is observed in rabbits immediately after birth. Subsequently, microbes slowly and continuously colonize the rabbit gut and the amount of antimicrobial substances in rabbit milk and nutrients have major effects on the development of gut functions and the microbial community [10,19]. A large variation in the bacterial community was observed between young rabbits at an early age. However, the composition of the bacterial community between rabbits became more similar with increasing age, especially with pelleted diets [13,19]. To our knowledge, the caecum is a major organ of bacterial fermentation in rabbits, providing around 50–60% of daily energy requirements. The caecal environment greatly affects the microbial community in rabbits, especially in terms of pH, being influenced in a major way by fermentable products called volatile fatty acids. A low nutrient fermentation efficiency can result in a higher value of caecal pH, increasing the risk of a digestive disorder [6]. The caecal pH of the RMR group seemed to be higher than that of the other groups. However, the caecal pH of the RMR group was lower than 6.73, insufficient to promote a gut health problem [6]. Moreover, there was no statistically significant difference in caecal pH, growth performance, digestibility, morbidity and mortality during the experiments, supporting the possibility of using RMR in rabbits.

Caecal enzyme activity was another indicator representing microbial activity. The amount of caecal amylase and protease in the RMR and KMR groups was higher than in the MMR group. Fermentation products from probiotic bacteria and prebiotics play an important role in supporting the development of a normal flora in rabbits [20]. Supplementation with fermentation products from several normal flora bacteria (*Lactobacillus* sp., *Enterococcus faecium*, *Bifidobacterium bifidum*, *Pediococcus adicilactici*) and prebiotics (fructooligosaccharide) in the KMR formulation contributed to higher enzyme activity, whereas RMR and MMR did not contain these supplements. Such findings are in accordance with another research study [8]. A higher survival rate was observed in the milk replacer with probiotics and prebiotics (35.3%) compared to without these additions (21.3%) in infant cottontail rabbits in which the chemical composition of these milk replacers were similar [8]. Thus, probiotics and/or prebiotics may have been the key factor determining this result. The unsuitable chemical components of KMR with respect to rabbit milk and the slow growth rate of bacteria in the first period could have been the cause of the lower feed intake in the first period in the KMR group. Subsequently, feed intake in the KMR group was higher than in the others at the end of the experiment, as a consequence of the full development of microbes. The closeness of the chemical component of RMR to that of rabbit milk may have promoted the appropriate substrate for early microbial colonization, leading to higher caecal enzyme activity. None of prebiotic supplement and difference in chemical composition in MMR could be the cause of the lower enzyme activity in the caecum compared to other groups. Therefore, supplementation with probiotics and/or substrates for a normal flora in feed formulation with the appropriate chemical components (i.e., in line with rabbit milk) may represent the best procedure to achieve good development of the microbial community in early weaned rabbits. However, a microbial community analysis should be performed in a future study to confirm this hypothesis.

No statistically significant difference in growth performance was observed in this study. However, a trend existed that can be explained in detail. The lowest FCR was observed in the RMR group as a consequence of the lower ADFI and the higher ADG. This may have been due to the appropriate crude protein and fat proportion in the RMR, which promoted lower adaptation following the milk replacer in the first period. However, the highest ADFI was observed in the KMR group as the lowest energy density because the feed intake was stimulated by a physiological function until the animal had obtained its daily energy requirement [6]. Therefore, a lower pelleted diet intake could be observed in the group already receiving a high-density milk replacer in the RMR and MMR groups. The digestibility of dry matter and nutrients in the RMR group tended to be lower than in the other groups. The supplementation of prebiotics and substrates for microflora in KMR can support the microbial community and facilitate increased digestibility. In addition, MMR contained the highest energy density, which bacteria can use as energy sources to produce butyrate. Intestinal epithelial cells can utilize such short-chain fatty acids as an energy source, promoting proliferation, differentiation and gut immunity [21]. Thereby, the greatest villus height and villus crypt depth were seen in the MMR group, enabling better digestion and absorption and providing higher nutrient digestibility RMR as a result [22]. Furthermore, the amount and characteristics of hard faeces can be used to indicate digestive health [7]. The total weight of faecal excretion did not differ between groups, but the largest number of faecal pellets was found in MMR, followed by KMR and RMR. Moreover, no soft faeces were found under the rabbit cage, indicating that crude protein intake did not exceed their requirement [19].

Lower growth performance and survival rates were reported in the group fed with KMR compared with the group fed with rabbit milk from lactating does [8]. Unfortunately, this study did not compare the milk replacers with rabbit milk. Early separation at 14 days of age may have been the cause of the negative consequences found in the study of [8], whereas separation at 18 days of age did not result in any serious adverse outcomes here. In addition, a high mortality rate was observed in the study of [8] because the rabbits were too young, were injured and were experiencing high levels of stress due to being wild rabbits. 

## 5. Conclusions

Differences in efficiency between the RMR developed in this study and commercial milk replacers (KMR and MMR) used in 18-day-old rabbits were revealed in the current study. Based on the results, it was possible to use RMR as a milk replacer for 18-day-old rabbits and to wean at 36 days of age, this not providing any adverse consequences for final body weight, ADG, FCR, ADFI, nutrient digestibility, internal organ characteristics, caecal pH, amount of faeces excretion and crude protein intake. Lower nutrient digestibility was observed in the RMR group without statistically significant differences. Therefore, probiotics and/or prebiotics can be supplemented in formulations to promote a suitable microbial community and to provide benefits in terms of growth performance and nutrient digestibility.

## Figures and Tables

**Figure 1 animals-10-01087-f001:**
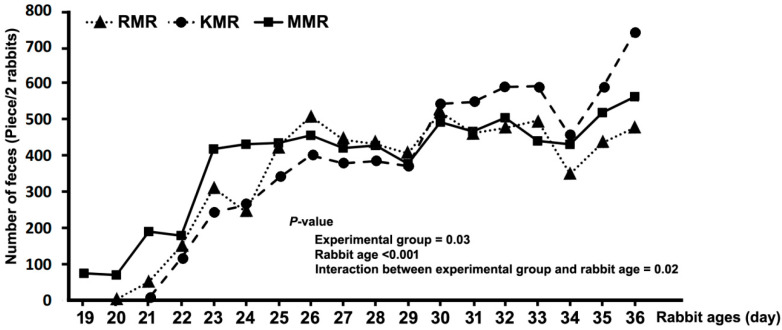
Effect of different milk replacers on number of faecal pellets (RMR, rabbit milk replacer in this study; KMR, kitten milk replacer, KMR^®^, Pet-Ag Inc., Hampshire, IL, USA; and MMR, mammal milk replacer, Zoologic^®^ Milk matrix 30/52, Pet-Ag Inc., Hampshire, IL, USA).

**Table 1 animals-10-01087-t001:** Ingredients and chemical components of different milk replacers, rabbit milk and diet.

Items	Artificial Milk Replacers			Rabbit Diet ^a^	Rabbit Milk ^b^
	RMR	KMR	MMR		
Ingredients (%)					
Sodium caseinate	43.8	-	-	-	-
Skimmed milk powder	7.73	-	-	-	-
Hydrogenated palm fat	41.5	-	-	-	-
Monodicalcium phosphate	3.13	-	-	-	-
Limestone	2.30	-	-	-	-
Salt	0.94	-	-	-	-
Premix ^c^	0.50	-	-	-	-
Polyoxyethylene (80) sorbitan monooleate	0.10	-	-	-	-
Chemical composition					
Dry matter (%FM)	95.8	96.2	95.2	90.7	29.8
Crude ash (%DM)	9.77	6.66	8.27	5.69	7.38
Crude protein (%DM)	41.4	48.5	30.1	16.1	41.3
Ether extract (%DM)	43.8	22.9	52.5	2.42	43.3
Crude fiber (%DM)	ND	ND	ND	25.2	ND
Nitrogen free extract (%DM) ^d^	5.03	22.0	9.13	50.6	8.05
Metabolizable energy content (kcal/100g DM) ^e^	535	441	584	254	540

RMR = Rabbit milk replacer which was performed in this study, KMR = Kitten milk replacer (KMR^®^, Pet-Ag Inc., Hampshire, IL, USA), MMR = Mammal milk replacer (Zoologic^®^ Milk matrix 30/52, Pet-Ag Inc., Hampshire, IL, USA), FM = Fresh matter, DM = Dry matter, ND = Not detect; ^a^ A commercial pelleted diet for rabbits (Lee Feed Mill, Publ. Co., Ltd., Phetchaburi, Thailand); ^b^ Chemical composition of rabbit milk [10]; ^c^ Vitamin and mineral premix (Topmix-B111, Top Feed Mills Co., Ltd., Pathumthani, Thailand) were supplied per kilogram of diets at 4,800,000 IU of vitamin A; 1,200,000 IU of vitamin D3; 6000 IU of vitamin E; 600 mg of vitamin K; 600 mg of vitamin B1; 2200 mg of vitamin B2; 10,000 mg of vitamin B3; 800 mg of vitamin B6; 4 mg of vitamin B12; 48 mg of biotin; 4800 mg of Calcium pantothenate acid; 200 mg of folic acid; 24,000 mg of Zn, 16,000 mg of Fe; 32,000 mg of Mn; 32,000 mg of Cu; 200 mg of I; 40 mg of Se; 40 mg of Co; ^d^ Calculation [12]; ^e^ Calculation based on Atwater system [13].

**Table 2 animals-10-01087-t002:** Effect of different milk replacers on rabbit performances, apparent digestibility and crude protein intake.

Parameters	Factors							SEM	*p*-Value		
	Artificial Milk Replacers (AMR)			Age (days)					AMR	Age	AMR * Age
	RMR	KMR	MMR	0	6	12	18				
BW (g/head)	210	198	202	134 ^a^	157 ^b^	214 ^c^	308 ^d^	9.086	0.38	0.001	0.94
				19–24	25–30	31–36	-				
ADFI (g/head/day)	22.1	26.7	22.1	7.47 ^a^	22.1 ^b^	39.6 ^c^	-	2.239	0.59	0.001	0.56
ADG (g/day)	9.62	9.33	8.52	4.40 ^a^	10.8 ^b^	14.6 ^c^	-	0.808	0.84	0.001	0.77
FCR	2.39	2.62	2.74	1.91 ^a^	2.21 ^a^	2.75 ^b^	-	0.115	0.44	0.01	0.56
Water intake (g/head/day)	41.8	30.9	34.2	10.1 ^a^	31.9 ^b^	68.0 ^c^	-	3.902	0.1	0.001	0.38
Faeces excretion (g/head/day)	15	14.3	15.4	1.29 ^a^	5.49 ^b^	8.14 ^c^	-	0.936	0.81	0.001	0.32
Crude protein intake (g/head/day) ^1^											
Diet	3.72	3.73	3.9	1.20 ^a^	3.56 ^b^	6.37 ^c^	-	0.361	0.59	0.001	0.56
Diet and milk	5.95	6.34	5.52	3.39 ^a^	5.65 ^b^	8.56 ^c^	-	0.364	0.11	0.001	0.56
				23–27	31–35	-	-				
Apparent digestibility (%)											
Dry matter	59.6	63.3	63.5	60.5	63.4	-	-	1.126	0.29	0.22	0.96
Organic matter	60.7	65.1	65.2	62.8	64.2	-	-	1.123	0.18	0.51	0.85
Ether extract	68.3	73.8	70.7	71.3	70.3	-	-	2.418	0.75	0.88	0.97
Crude protein	76.5	80.8	81.1	81.5 ^b^	77.1 ^a^	-	-	1.007	0.07	0.03	0.34

RMR = Rabbit milk replacer, which was used in this study, KMR = Kitten milk replacer (KMR^®^, Pet-Ag Inc., Hampshire, IL, USA), MMR = Mammal milk replacer (Zoologic^®^ Milk matrix 30/52, Pet-Ag Inc., Hampshire, IL, USA), SEM = pooled standard error of mean, BW = Body weight, ADFI = Average daily feed intake, ADG = Average daily weight gain, FCR = Feed conversion ratio; ^a, b, c, d^ The differences in superscript letter in the same row represented statistical significant differences (*p* < 0.05); ^1^ The rabbits in the RMR, KMR and MMR groups received crude protein from milk at 2.23, 2.61 and 1.62 g/head/day, respectively.

**Table 3 animals-10-01087-t003:** Effect of different milk replacers on internal organ weight, caecal pH, intestinal villi morphology and digestive enzyme activity.

Parameters	Artificial Milk Replacers (AMR)			SEM	*p*-Value
	RMR	KMR	MMR		
Internal organs characteristics (g/live body weight)					
Liver	4.46	4.00	3.81	0.162	0.25
Spleen	0.14	0.10	0.10	0.015	0.51
Kidney	1.24	1.18	1.12	0.046	0.61
Thoracic organs ^1^	1.07	1.09	1.15	0.063	0.38
Pancreas	0.06	0.07	0.05	0.007	0.99
Full stomach	7.40	7.44	7.22	0.555	0.79
Stomach wall	2.09	2.01	1.99	0.061	0.65
Intestinal organs ^2^	7.38	6.87	7.37	0.242	0.54
Full caecum	16.3	16.1	16.2	0.422	0.97
Caecal wall	2.75	2.49	2.34	0.134	0.48
Caecal pH	6.50	6.30	6.38	0.081	0.63
Caecal digestive enzyme activities (U)					
Amylase	12.2 ^ab^	18.5 ^b^	10.4 ^a^	1.41	0.04
Protease (×10^−1^)	7.26 ^b^	6.84 ^ab^	5.79 ^a^	0.253	0.04
Cellulase	3.71	3.50	3.60	0.057	0.31
Duodenal villi morphology (µm)					
Villus height	320 ^a^	361 ^ab^	380 ^b^	8.50	0.04
Villus crypt	66.2 ^ab^	62.6 ^a^	68.4 ^b^	0.737	0.007

RMR = Rabbit milk replacer, which was used in this study, KMR = Kitten milk replacer (KMR^®^, Pet-Ag Inc., Hampshire, IL, USA), MMR = Mammal milk replacer (Zoologic^®^ Milk matrix 30/52, Pet-Ag Inc., Hampshire, IL, USA), SEM = pooled standard error of mean, BW = Body weight, FI = Feed intake, ADG = Average daily weight gain, FCR = Feed conversion ratio; ^a, b^ The differences in superscript letter in the same row represented statistical significant differences (*p* < 0.05); ^1^ Thoracic organs includes lungs and heart; ^2^ Intestinal organs includes stomach, small intestine, large intestine, caecum and rectum with content.

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
