# Peer review of "Comparison of Rabbit, Kitten and Mammal Milk Replacer Efficiencies in Early Weaning Rabbits"

_animals, 2020, doi:10.3390/ani10061087_

Round 1
Reviewer 1 Report
Dear Editor and authors,
The manuscript content is interesting however some doubts should be explained. You can find my suggestions below and highlighted in manuscript.
- If authors compared in their study RMR with another commercial products like KMR and MMR I suggest to change the title:
Comparison of rabbit, kitten and mammal milk replacers efficiency in early-weaning rabbits.
- Abstract
line 37-38
…observed in growth performance parameters, water intake….
Line 39
Amylase was signif. higher only in KMR vs MMR and protease in RMR vs MMR Tab.3
Line 40
Villus height was signif. higher only in MMR vs RMR and villus crypt in MMR vs KMR Tab. 3
Line 42-43
From Table 2 digestibility parameters were not significantly affected
- Introduction
Line 48-56
Why authors describe the situation with pet only in England? Do you have the special reason?
- Materials and methods
Line 85
How did you present the results, as standard deviation or standard error (in tables)?
Statistical analyses
All parameters except Table 3 are compared between different milk replacers and also in different age period, so one factor is treatment and the second one is age. Why did you use for statistical analysis two way ANOVA only for number of faeces pellets? If the interaction between treatment x age is significant you can use the simple one-way analysis of variance with post hoc Tukey (Duncan) test. Moreover, in this section you should describe how the data are presented (mean values / SD or SEM).
Please use the two way analysis for all parameters or explain why did you decide to evaluate the data by one way analysis.
- Results
Table 2
I suggest the title of table: Effect of different milk replacers on rabbit performance.
You should insert another table only with parameters of digestibility.
Please define in footnote SEM (or SD) and P value (same for all tables)
Line 178-179
Amylase was signif. higher only in KMR vs MMR and protease in RMR vs MMR
Villus height was signif. higher only in MMR vs RMR and villus crypt in MMR vs KMR
Line 184
I suggest the title of table: Effect of different milk replacers on internal organs weight, caecal pH, intestinal villi morphology and digestive enzyme activity.
Table 3:
Please explain how did you express the internal organs weight (g/body weight ?).
Specify thoracic organs, intestinal organs
Stomach and caecal wall-do you mean empty stomach and caecum?
Please define the superscripts, pay attention you have same letters for differences between milk replacers and significants.
Fig. 1
I suggest the title of fig.1 : Effect of different milk replacers on number of faecal pellets.
If the interaction between treatment and age was significant you have to calculate significant differences (Tukey/Duncan).
37th day is missing in graph
- Discussion
Line 319-320
Please, shortly explain how can butyric acid affect intestinal villus height. Which is the mode of action?
- Conclusion
This section is too long. Please describe more clearly the main result you obtained and formulate your own opinion and suggestion.

Author Response
Dear reviewer
We carefully corrected the manuscript following your suggestions as in attached file.
If you have any suggestion or questions, please ask us. We would like to correct and explain to you.
Best regards
Attawit Kovitvadhi

Reviewer 2 Report
In this work Chankuang et al. formulated rabbit milk replacer (RMR), which possible to fed early weaning rabbits without serious adverse effects. English must be edited by a native English speaker or by a professional English editor. Text is hard to follow in many parts due to an awkward syntaxis. Thus, interpretation of results and discussion can't be evaluated properly. After a resubmission, I will be able to judge more accurately the hypothesis and ideas proposed within the whole MS. The ANIMALS`S guideline to authors state that "For editors and reviewers to accurately assess the work presented in your manuscript you need to ensure the English language is of sufficient quality to be understood.". As said in this statement, the English does not have to be perfect, but sufficient to correctly interpret the results. From reading the manuscript, the quality of English does not seem sufficient to me, and I therefore suggest improving the English of the manuscript.
Introduction is out-of-scope with confusing, scientifically wrong and unclear. The hypothesis and research questions are not well presented in introduction; therefore L48-66 must be rephrasing accordingly.
Material and methods needs to describe the details in briefly of each experiment to be accurate evaluation how authors reached to findings.
The manuscript must be substantially improved according to the comments below:
L50: Why use the abbreviate “CAGR”?; which mentioned only one time throughout manuscript.
Fig. 2 should present as table to be clear means of cured protein intake at different age stage with SEM for the each period.
L81-84: Should separate under subtitle of Ethics statement.
L84-90: the statement is very long including more than two adverbs.
What is the procedures of “rabbit milk replacer RMR”
L91: “one rabbit per replicate” mention how many animals per group
L99: how much dry matter of replacer milk were intake per kg of body weight?
L135: replace “evaluation on villus morphology” with “villus morphometric examination”, the procedure should describe in briefly.
L140: replace “to study” with “to measure the content of”, the procedure should describe in briefly.
L148: extend “U” to the full name
L149: delete “Calculation and”
L150: “The daily crude protein intake was calculated at 19–24, 25–30 and 31–36 days of age based on feed and milk replacer intake (Figure 2), which was compared with the data of [17]” should be moved in in independent subtitle “crude protein assessment” before statistical analysis , the procedure should describe in briefly.
L145: replace “evaluated” with analyzed
L154: the variable faeces pellets should analysis using repeated measurement with include the time of collection in the model and the days of collection should divided as interval periods for example “3 or 4 days “ to present the significant findings. Fig. 1 convert to table, including groups mean, SEM, and p value.
Which variables analyzed by one-way and which variables analyzed by two-way ANOVA,
Duncan’s superscript letters should insert among significant means in Tables
Where is the section of morphometric analysis? should be presented in figure in results chapter
Author Response
Dear Reviewer
We carefully corrected the manuscript following your suggestions as in the attached file.
If you have any suggestion or questions, please ask us. We would like to correct and explain to you.
Best regards,
Attawit Kovitvadhi

Reviewer 3 Report
This manuscript presents the study on three different milk replacers (i.e., kitten-, mammal- and artificial rabbit milk replacers) to rabbit milk and the impact of three different milk replacers on the live performance, nutrient digestibility and gut histology in rabbits.
Major comments:
The authors did the digestibility trial following the protocol of reference [13]. However, there is an official document with an internationally recognized protocol for digestibility trials in the rabbit: European reference method for in vivo determination of diet digestibility in rabbits. Perez et al., 1995. World Rabbit Science Journal. The authors should have followed the above-mentioned protocol, where the minimum number of animals (replicates) per treatment at the end of the digestibility trial is 7 (and not 6 that is the number of replicates that authors used). Considering that the authors did not follow that protocol, they should describe accurately the protocol used.
Simple Summary: authors should rephrase the first sentence. As it is written, it seems that this is a common practice. Moreover, in lines 18-21 (and in introduction section lines 60-63) authors mention that this procedure promotes a rapid production cycle and it is beneficial to meat producers. In the opinion of the reviewer, this is not true. In commercial rabbit meat farms this practice is not used, for many reasons, of whose the most relevant is its cost. Authors should direct the interest of using milk replacers in pet rabbits, only. Modify accordingly along with the manuscript.
The caecal amylase activity of RMR was not different from that of MMR (line 39). Same for Protease activity. Along with the text, authors must pay attention that when superscript letters do not differ, there is no statistical difference.
Lines 130-131: for digestibility trial, the faeces must be weighed carefully, this is not mentioned in M&M section. Why authors took photos of the faeces (why they count the faeces and how? were all hard faeces)? In the opinion of the reviewer the hard faeces size changes not only according to the animals’ age, but also on the animal’s size, individual variation, and health status, so it does not represent a good measurement for any calculations. If in the digestibility trial the authors used as the output measurement the hard faeces number, this is not correct. Input and output must be measured with the same measurement unit, that is the weight unit. Thus, if the digestibility of nutrients was calculated as above, in the opinion of the reviewer it represents a weak point. Moreover, Figure 1 should be deleted.
English language must be revised by a native English speaker.
Minor comments:
In the abstract and result section “P” not in uniform. Please revise throughout the manuscript.
Line 49: Why did the authors refer to England? not other countries? Pet ownership is higher in many other countries than in England.
In the material and methods section: Authors never mentioned, how rabbits were housed and in what environmental conditions? Please revise it.
Line 96-96: What was the temperature of milk when it was provided to rabbits?
Line 111-123; 171-173; 186-188: In Table 2 & 3, delete the heading “Groups” and insert “Artificial milk replacers” like Table 1. Footnotes should be in one paragraph and separated by semicolon or comma or full stop (follow the manuscript guidelines).
Line 129 If authors didn’t follow the European reference method for in vivo diet digestibility in rabbits (Perez et al., 1995), they must describe the whole procedure.
Table 1: chemical composition and instead of proximate analysis; Metabolized is not correct, replace with: (metabolizable energy content kcal/100 g DM)
Line 137: -20 °C please revise it
Along with the manuscript: units must be separated from the number.
Line 169; 184: Table 2 & 3, delete heading “Groups” and insert “Artificial milk replacers” like Table 1.
Line 185: Table 3…….milk replacers
Line 199; 211: Delete extra inserted lines.
Line 203-210: Format paragraph spacing.
Line 234: replace “is” by “in”.
Line 304: Conclusions, authors repeating result section in conclusion, please rewrite the conclusion.
Line 356: Delete “For research articles with several authors, a short paragraph specifying their individual contributions must be provided. The following statements should be used”.
Line 372: Reference section: Volume numbers are missing.
For example: “Morley, C.; Fook, J. The importance of pet loss and some implications for services. Mortality 2005, 10, 127- 143”, should be “Morley, C.; Fook, J. The importance of pet loss and some implications for services. Mortality 2005, 10(2), 127- 143”.
The authors should add this reference: Comparison of outcomes using two milk replacer formulas based on commercially available products in two species of infant cottontail rabbits. Paul, G., Friend, D.G. 2017 Journal of Wildlife Rehabilitation 37(1), pp. 13-19
Author Response
Dear Reviewer
We carefully corrected the manuscript following your suggestions as in the attached file.
This paper has been professionally proofread by professionally company.
If you have any suggestion or questions, please ask us. We would like to correct and explain to you.
Best regards,
Attawit Kovitvadhi

Round 2
Reviewer 2 Report
The authors processed the comments and the manuscript became suitable for publication in the revised version
Author Response
Dear Reviewer
Thank you for your suggestion. You can find the revised the manuscript in the attached file.
Best regards,
Attawit Kovitvadhi

Reviewer 3 Report
Parentheses, solidus, percentage, dash/hyphen: font size and style are different, please revise it throughout the manuscript.
The authors made the changes according to the comments of the reviewer, but still further changes are required in the manuscript:
Line 130: feed intake was collected? please revise it.
In table 3: Change Artificial milk replacersa to Artificial milk replacers (AMR)
Line 250: Delete hyphens.
Line 364-365: Delete, “please turn to the CRediT taxonomy for the term explanation. Authorship must be limited to those who have contributed substantially to the work reported”.
Author Response
Dear Reviewer
Thank you for your suggestion. You can find the revised the manuscript in the attached file. The details of correction in revised manuscript was presented below following your suggestion.
Best regards,
Attawit Kovitvadhi
